# Smoking Cessation Apps: A Systematic Review of Format, Outcomes, and Features

**DOI:** 10.3390/ijerph182111664

**Published:** 2021-11-06

**Authors:** María Barroso-Hurtado, Daniel Suárez-Castro, Carmela Martínez-Vispo, Elisardo Becoña, Ana López-Durán

**Affiliations:** 1Smoking and Addictive Disorders Unit, University of Santiago de Compostela, 15782 Santiago de Compostela, Spain; danielsuarez.castro@usc.es (D.S.-C.); carmela.martinez@usc.es (C.M.-V.); elisardo.becona@usc.es (E.B.); ana.lopez@usc.es (A.L.-D.); 2Department of Clinical Psychology and Psychobiology, University of Santiago de Compostela, 15782 Santiago de Compostela, Spain

**Keywords:** smoking cessation, mHealth, smartphone, mobile phone, digital health, systematic review

## Abstract

Smoking cessation interventions are effective, but they are not easily accessible for all treatment-seeking smokers. Mobile health (mHealth) apps have been used in recent years to overcome some of these limitations. Smoking cessation apps can be used in combination with a face-to-face intervention (FFSC-Apps), or alone as general apps (GSC-Apps). The aims of this review were (1) to examine the effects of FFSC-Apps and GSC-Apps on abstinence, tobacco use, and relapse rates; and (2) to describe their features. A systematic review was conducted following the internationally Preferred Reporting Items for Systematic Reviews and Meta-Analyses (PRISMA) statement. Of the total 6016 studies screened, 24 were included, of which nine used GSC-Apps and 15 FFSC-Apps. Eight studies reported significant differences between conditions in smoking cessation outcomes, with three of them being in favor of the use of apps, and two between different point-assessments. Concerning Apps features, most GSC-Apps included self-tracking and setting a quit plan, whereas most of the FFSC-Apps included self-tracking and carbon monoxide (CO) measures. Smartphone apps for smoking cessation could be promising tools. However, more research with an adequate methodological quality is needed to determine its effect. Nevertheless, smartphone apps’ high availability and attractiveness represent a great opportunity to reach large populations.

## 1. Introduction

Smoking is the main avoidable cause of morbidity and mortality worldwide [1]. Tobacco components are related to different harmful cardiovascular and respiratory effects [2]. Specifically, the more common physical illnesses related to smoking are cancer, cardiovascular diseases, respiratory diseases, and reproductive problems [3]. Moreover, tobacco use causes around 8 million deaths every year [4]. Smokers also have a poorer quality of life [5] and a higher likelihood of having mental health problems [6], such as depression or anxiety [7].

The U.S. Preventive Services Task Force Recommendation Statement proposes that behavioral interventions are effective for smoking cessation in adults, adolescents, and pregnant women [8]. However, traditional treatments have limitations such as low utilization [9,10], they do not provide fast and tailored responses to smoking urges [11], they require costly resources and health services [12], and smoking cessation services are not easily accessible to all treatment-seeking smokers [13].

Some of these limitations could be minimized or eliminated with the use of ICT (Information and Communication Technology), as it can provide support for quitting smoking through Short Message Service (SMS) [14,15] or smartphone apps [16]. In fact, the relevance and usage of mHealth (mobile Health), defined as the use of a Personal Digital Assistant (PDA), mobile phones, wireless devices, and monitoring devices for clinical practice [12], to quit smoking is increasing.

mHealth smartphone apps targeting smoking cessation have undergone significant development and increases in recent years due to some characteristics such as: (a) they can be used anywhere at any time; (b) they are cost-effective; (c) they can send tailored messages according to user characteristics; (d) they can deliver different messages according to time and location; (e) they can offer support for tobacco cravings; and f) they can provide social support [17]. Smoking cessation smartphone apps could be classified as: (1) general smartphone apps for smoking cessation without face-to-face contact (GSC-Apps); or (2) smoking cessation smartphone apps combined with face-to-face intervention (FFSC-Apps) [17]. Both kinds of smartphone apps could improve smoking cessation efforts in different ways. GSC-Apps enable widespread distribution to people who do not have access to face-to-face treatment, and they could be better tailored than a face-to-face treatment [16]. Furthermore, GSC-Apps could be a very useful option for people who do not wish to undergo face-to-face treatment. Although the results about its effectiveness are scarce, and the cost and benefit of this kind of treatment must always be considered. Compared to face-to-face treatment, smoking cessation smartphone apps could increase the intensity of the behavioral treatment, as different studies have shown that high-intensity smoking cessation interventions are more effective than low-intensity interventions [18,19]. Therefore, both kinds of smartphone apps have different aims: GSC-Apps facilitate access to tools to quit smoking to a greater number of people, and FFSC-Apps could complement face-to-face interventions. In this vein, adding a smartphone app as a complement to a face-to-face treatment might improve abstinence outcomes.

As over 3.6 billion people have smartphones [20], and more than 204 billion mobile apps have been downloaded in 2019 worldwide [21], the use of smartphone apps provides an opportunity to cope with difficulties in providing smoking cessation treatment and improving its effectiveness. Despite the great availability of smartphone apps in the health field, only a few have reported information about their quality and reliability [12]. In this vein, some studies have concluded that few smartphone apps follow the recommendations of the smoking cessation clinical guidelines [22,23,24,25]. For instance, Haskins et al. [26] found that only two of the 50 smartphone apps recommended by the app store had scientific support. In the same line, Vilardaga et al. [16], in a systematic review analyzing smoking cessation smartphone apps, suggested that a greater effort to connect app features with clinical outcomes is needed. Moreover, there are no studies that have examined the effects and features of smartphone apps combined with face-to-face intervention and smartphone apps without face-to-face contact. In light of the reviewed literature, specific information is needed about the characteristics of the smartphone apps for smoking cessation and the advantages and disadvantages of the different technological formats used. Thus, the aims of this study were (1) to systematically review the literature that has explored the effect of smartphone apps for smoking cessation (combined with face-to-face intervention or general apps for smoking cessation) on abstinence, tobacco use, and relapse rates; and (2) to describe the features of smartphone apps for smoking cessation.

## 2. Materials and Methods

### 2.1. Search Strategy

This systematic review followed the Preferred Reporting Items for Systematic Reviews and Meta-Analysis (PRISMA) statement [27], and the review protocol was registered with PROSPERO (CRD42020154272), an international database for the registration of systematic reviews in the field of health. Databases searched were Pubmed and PsycINFO. The complete literature search strategy can be found in Appendix A. Additionally, a search of the first 200 citations published online in Google Scholar was undertaken.

We included studies published in English and Spanish, and all the years available in the selected databases, with the limit of 30 November 2020.

### 2.2. Study Selection Criteria

#### 2.2.1. Study Characteristics

We included the following study designs: (a) experimental studies (randomized controlled trials, quasi-randomized trials, and controlled clinical trials), (b) quasi-experimental studies (interrupted time series and before-and-after studies), and (c) observational studies (cohort studies, case-control studies, and case series).

Qualitative studies, research protocols, and review studies were excluded.

#### 2.2.2. Participants

The population included in this systematic review were adult daily smokers (aged 18 and over). For the purposes of this review, daily smokers were defined depending on the study criteria. The definition of a daily smoker is indicated in the description of each study (Appendix A).

#### 2.2.3. Type of Intervention

Included studies were those examining smartphone apps for smoking cessation whose aim was to quit smoking. These apps were classified into the following categories: (1) general apps for smoking cessation, which do not include face-to-face health professional contact (GSC-Apps); or (2) smoking cessation apps combined with face-to-face intervention (FFSC-Apps).

#### 2.2.4. Exclusion Criteria

Excluded studies were those in which: (1) participants had cognitive impairment or a substance use disorder (e.g., alcohol, cannabis, and opioid), or (2) participants were pregnant women. Participants with cognitive impairment were excluded due to the difficulties of understanding the intervention content, while participants with substance use disorders were excluded due to the influence that substance use has on achieving and maintaining tobacco abstinence [28]. Finally, pregnant women were excluded because their motivation to quit is different from the general population, and the content of apps targeting this population is tailored to their particular circumstances [29].

### 2.3. Outcomes

The primary outcome was the effect of smartphone apps for smoking cessation on tobacco use, abstinence, and relapse rates.

The secondary outcome was to describe the features of smartphone apps for smoking cessation.

### 2.4. Study Selection

First, titles and abstracts retrieved by electronic searches were exported to a reference management software (Sciwheel) to remove duplicates. These references were then exported to the online software tool Abstrackr for screening. This is a free tool to upload and organize the results obtained in systematic reviews, allowing one to filter and organize the abstracts of the articles on a single platform. Reviewers DSC and MBH screened titles and abstracts independently. These reviewers discussed disagreements, and other reviewers (CMV, EBI, ALD) were involved if a decision was not reached. Full-text screening, data extraction, and quality assessment were performed independently by both reviewers (DSC and MBH). Excluded studies, with the reasons, are recorded in the PRISMA flow diagram (Figure 1).

### 2.5. Data Extraction and Analysis

Data were extracted independently by DSC and MBH using an electronic data extraction form of Microsoft Excel 2016. Extracted information included: the study’s general information, study design, participant characteristics, sample size, method of assessment, kind of intervention, group size for group-based intervention, app’s features, components of the intervention and control conditions, and tobacco use outcomes.

If necessary, the primary authors of the studies were contacted and asked to provide any additional or missing data.

### 2.6. Assessment of Risk of Bias

Two reviewers (DSC and MBH) assessed the quality of the studies that met the eligibility criteria for this review. The quality assessment was performed using the Effective Public Health Practice Project Quality Assessment Tool (EPHPP). This tool has been deemed suitable for a systematic review of effectiveness [30]. The EPHPP tool assesses six domains: (a) selection bias regarding whether the participants are representative of the target sample and how many agreed to participate; (b) study design relating to the probability of bias due to the allocation process; (c) confounders specifying possible differences before the intervention and, if applicable, if the confounders were controlled; (d) blinding refers to whether the study blinded the outcome assessor and blinded the participants; (e) data collection method defined as the tools for the outcome measures being described as valid and reliable; and (f) withdrawals/dropouts regarding whether the study specifies the numbers and reasons for withdrawals/dropouts and the percentage of participants who completed the study [31]. Each study receives a global quality score once the specific domains are assessed (strong, moderate, or weak). Furthermore, this tool assesses two extra dimensions: (a) intervention integrity (the number of participants who receive the intervention, its consistency, and if they receive any unintended intervention that may influence the results); and (b) analysis appropriate to question (whether the quantitative analysis was appropriate for the research question).

## 3. Results

A total of 6016 studies were screened after duplicates were removed. Of these, 5888 were excluded after the review of titles and abstracts. In the second phase, 128 papers were read in full text. After evaluation of the full report, 104 studies were excluded. Finally, 24 publications were included in this review.

### 3.1. Study Characteristics

The characteristics of the studies included are described in detail in Appendix A. Of the 24 included studies, 19 (79.1%) were conducted in the United States, two (8.3%) in Japan, one (4.2%) in Israel, one (4.2%) in Ireland, and one (4.2%) in Australia. Regarding the study type, nine (37.5%) were RCTs e.g., [32,33], four (16.7%) were controlled clinical trials (CCTs) [34,35,36,37], and eleven (45.8%) were before-and-after e.g., [38,39]. Fifteen (62.5%) used FFSC-Apps e.g., [39,40] and nine (37.5%) used GSC-Apps e.g., [32,33]. Regarding the studies with FFSC-Apps, two (13.3%) of these studies were pilot [37,41] and two (13.3%) were feasibility studies [35,36]. Regarding GSC-Apps, two (22.2%) of these studies were initial evaluations [42,43]. Fifteen studies (62.5%) included intervention assessment points ranging from 1 to 52 weeks, and four studies (16.7%) included follow-up assessments ranging from post-treatment to 6 months.

### 3.2. Methodological Quality Assessment

According to the EPHPP Tool, six studies received a quality rating of strong [32,33,41,44,45,46], thirteen studies of moderate e.g., [34,38,39], and five of weak [36,40,42,47,48]. Study design and withdrawal dimensions were the main strengths of the included studies, whereas the confounder dimension was the main weakness. The specific and global methodological quality ratings of each study are described in Table 1.

In addition, the EPHPP tool assesses two extra dimensions, intervention integrity and analysis appropriate to question. These dimensions are also included in this review. Regarding the intervention integrity, thirteen studies scored in the 80–100% category [34,35,36,39,41,43,45,46,49,50,51,52,53], four in the 60–79% [40,44,48,54], two in the less than 60% [42,47], and five did not provide this information [32,33,37,38,55]. Concerning the analysis component, the unit of analysis and allocation were individuals except for Raiff et al. [48], where it was pairs of smokers. All studies used appropriate statistical methods (based on the dimensional analyses of the EPHPP tool, which measure if the statistical methods are appropriate for the study design), and thirteen studies used an intention-to-treat approach [32,35,36,37,40,41,42,43,45,46,47,50,55].

**Table 1 ijerph-18-11664-t001:** Ratings of methodological quality by the EPHHP tool.

	Selection Bias	Study Design	Confounders	Blinding	Data Collection	Withdrawals	Global Rating
Baskerville et al. (2018) [55]	Strong	Strong	Strong	Strong	Strong	Weak	Moderate
BinDhim et al. (2018) [32]	Strong	Strong	Strong	Strong	Strong	Strong	Strong
Bricker et al. (2014) [33]	Strong	Strong	Strong	Strong	Strong	Strong	Strong
Bricker et al. (2017) [38]	Moderate	Moderate	Weak	Moderate	Strong	Strong	Moderate
Buller et al. (2014) [44]	Moderate	Strong	Strong	Strong	Strong	Moderate	Strong
Businelle et al. (2016) [39]	Moderate	Moderate	Weak	Moderate	Strong	Moderate	Moderate
Carpenter et al. (2015) [40]	Weak	Moderate	Weak	Moderate	Strong	Moderate	Weak
Dan et al. (2016) [54]	Moderate	Moderate	Weak	Moderate	Strong	Moderate	Moderate
Dar (2017) [34]	Moderate	Moderate	Weak	Moderate	Strong	Strong	Moderate
Garrison et al. (2020) [47]	Weak	Strong	Weak	Strong	Strong	Weak	Weak
Hébert et al. (2020) [41]	Strong	Strong	Strong	Moderate	Strong	Moderate	Strong
Hertzberg et al. (2013) [35]	Weak	Strong	Strong	Moderate	Strong	Strong	Moderate
Hicks et al. (2017) [36]	Weak	Strong	Weak	Moderate	Strong	Weak	Weak
Iacoviello et al. (2017) [43]	Strong	Moderate	Weak	Moderate	Strong	Strong	Moderate
Janes et al. (2019) [49]	Weak	Strong	Strong	Strong	Strong	Strong	Moderate
Krishnan et al. (2019) [37]	Strong	Strong	Weak	Moderate	Strong	Moderate	Moderate
Marler et al. (2019) [42]	Weak	Moderate	Weak	Moderate	Strong	Strong	Weak
Masaki et al. (2019) [50]	Strong	Moderate	Weak	Moderate	Strong	Strong	Moderate
Masaki et al. (2020) [46]	Moderate	Strong	Strong	Moderate	Strong	Strong	Strong
McClure et al. (2018) [51]	Moderate	Moderate	Weak	Moderate	Strong	Strong	Moderate
Minami et al. (2018) [52]	Strong	Moderate	Weak	Moderate	Strong	Strong	Moderate
O’Connor et al. (2020) [45]	Strong	Strong	Strong	Moderate	Strong	Strong	Strong
Raiff et al. (2017) [48]	Weak	Moderate	Weak	Moderate	Strong	Weak	Weak
Wilson et al. (2019) [53]	Moderate	Moderate	Weak	Moderate	Strong	Moderate	Moderate

### 3.3. Effects of Smartphone Apps on Abstinence, Tobacco Use, and Relapse Rates

#### 3.3.1. General Apps for Smoking Cessation (GSC-Apps)

Nine studies included GSC-Apps (Appendix A). Of these studies, six used comparison groups, and three were before-and-after studies.

Of the six studies that used comparison groups, five were RCTs [32,33,44,47,55] and one was CCT [34]. Of these, three compared two apps [32,33,47], one compared a GSC-App to a text messaging system [44], one compared a smartwatch and app to a wait-list control condition (participants of the control group only filled out the baseline questionnaire and did not receive a smartwatch) [34], and one compared a Crush the Crave (CTC) app to an On the Road to Quitting (OnRQ) print-based self-help guide [55]. Concerning smoking cessation outcomes (Table 2), two studies found significant differences in abstinence rates between conditions at each point-assessment [32,44] and one at the 6-months but not at the 3-months point-assessment [55]. The remaining two studies reported similar quit rates between the experimental and the control group at each point-assessment [33,47]. Regarding cigarettes per day (CPD) outcomes, one study found significant differences between conditions [34] and two studies did not find significant differences [47,55].

Regarding the three before-and-after studies, two obtained similar quit rates at the 30-day point prevalence abstinence at the end of the treatment, with 26.2% and 27.6%, respectively [42,43], while Bricker et al. [38] reported that 11% of the participants were abstinent at the 2-month follow-up. Regarding CPD outcomes, two studies found reductions over time [38,42].

Concerning relapse rate outcomes, none of the included GSC-apps studies provided such information.

#### 3.3.2. Combine Apps with Face-to-Face Contact (FFSC-Apps)

Face-to-face contact is defined as at least one visit to the laboratory or to the smoking cessation service. In this review, fifteen studies included FFSC-Apps (Appendix A). Of these studies, seven used comparison groups, and eight were before-and-after studies.

Of the seven studies that used comparison groups, four were RCTs [41,45,46,49] and three were CCTs [35,36,37]. Four studies compared two mobile apps [35,36,46,49], one compared the use of a mobile app (experimental group) to brief advice (control group) [37], and two studies compared three treatment conditions [41,45]. Regarding smoking cessation outcomes (Table 2), one showed significant differences in abstinence rates between groups [46]. Regarding CPD outcomes, one study found significant reductions in CPD from baseline to the 1-month follow-up [49], and one study showed that participants who had not stopped smoking in the combined condition (app combined with Acceptance and Commitment Therapy (ACT) face-to-face treatment) reported significantly less CPD at post-treatment compared to the other two conditions. Finally, one study found no significant differences between study arms [37]. The remaining studies did not report CPD outcomes.

Regarding the eight before-and-after studies (see Appendix A), two studies showed significant differences in terms of abstinence between point-assessments [39,48]. The remaining studies did not analyze point-assessment differences. Finally, two studies reported reductions in CPD [52,53].

Concerning relapse outcomes, one study found significant differences in the time to the first lapse after the quit date, being significantly higher in the experimental group than in the control group [46].

### 3.4. Effects of Smartphone Apps on Abstinence, Tobacco Use, and Relapse Rates Regarding the Methodological Quality of the Studies

Regarding methodological quality, six studies obtained a strong quality in the EPHPP tool [32,33,41,44,45,46]. All of them (6/6) measured abstinence outcomes [32,33,41,44,45,46], one of them (1/6) also measured CPD outcomes [45], and one (1/6) also measured relapse rate outcomes [46]. In addition, all of them used active comparison groups (e.g., apps, SMS). Of these studies, two obtained significantly higher abstinence rates in the experimental group compared to the control group [32,46], and one found significantly higher abstinence rates in the control group compared to the experimental group [44] (see Figure 2). Moreover, the study that also measured CPD [45] showed significantly higher reductions in CPD in the experimental group. In the same vein, the study that also measured relapse outcomes [46] obtained a significantly higher time to the first lapse after the quit date in favor of the experimental group compared with the control condition.

Of these six strong quality studies, three used GSC-apps [32,33,44] and three used FFSC-apps [41,45,46].

### 3.5. Features of Smartphone Apps for Smoking Cessation

Regarding the features of smoking cessation apps (Table 3), we clustered them into the following groups depending on the content of the apps: CO; set a quit date; EMAS; self-tracking or smoking self-report; mindfulness content; and ACT content. Specific Apps features reported in each study are shown in Appendix A.

Of the fifteen FFSC-Apps studies, thirteen included self-tracking/smoking self-reports, ten included carbon monoxide (CO) measures, four set a quit plan/quit date, four showed EMAS, two presented mindfulness content, and one included ACT content. Of the nine GSC-Apps studies, eight included self-tracking/smoking self-reports, six set a quit plan/quit date, two included ACT content, one used app CO measures, and one presented mindfulness content. A comparison of features of FFSC-Apps and GSC-Apps is shown in Figure 3.

The specific features and components of the different apps for smoking cessation are summarized in Table 4. The Kakao Talk [48] app was not included in Table 4, because it is an online social support forum app, not complying with any of the established categories.

## 4. Discussion

Overall, the results obtained in this review suggest that smoking cessation apps are useful tools for smoking cessation. Most studies using a comparison group showed that smartphone apps were at least as useful as the control conditions (e.g., brief advice, other mobile apps), obtaining abstinence rates at the end of treatment ranging from 36 to 100%. Regarding before-and-after studies, the abstinence rates obtained ranged between 12.5 and 51.5%. Despite these abstinence outcomes being lower than those obtained in conventional psychological and pharmacological interventions [56], the possibility of increasing treatment access to a wider population of smokers makes them promising tools in terms of public health impact. Additionally, results from studies measuring CPD suggest that smoking cessation apps are also as effective as control groups (e.g., print-based self-help materials, other mobile apps) in reducing cigarette use. More research is needed to obtain more accurate conclusions about relapse rates, because only one study assessed this outcome.

When considering the methodology quality of studies, six rated as strong. Half of these studies did not show significant differences in abstinence rates between conditions [33,41,45]. Regarding other smoking cessation outcomes (CPD and relapse rates), one study measured CPD at post-treatment [45] and another study assessed relapse rates [46], finding both significant data in favor of the experimental group.

Regarding apps classification, in those studies using GSC-Apps, two obtained significantly better smoking cessation outcomes [32,34] in the experimental group compared to the control group. Two studies found better abstinence rates in the control group [44,55], and one study found a significant reduction in CPD in both conditions [47]. Concerning FFSC-Apps studies, two obtained significantly better smoking-related outcomes in favor of the experimental condition [45,46], compared to the control condition. One study found significant reductions in CPD in both conditions [49]. When examining studies without a comparison group (before-and-after studies), two showed significant differences in abstinence rates at the different point-assessments, being higher compared to the baseline assessment [39,48]. Finally, only one study [46] analyzed relapse outcomes, finding a significantly higher time to the first lapse after the quit date in the experimental group.

The following factors could influence the findings of this systematic review. First, seven studies targeted specific populations using apps for smokers with posttraumatic stress disorder [35,36], smokers with schizophrenia [53], smokers with mood disorders [52], homeless veteran smokers [40], smokers diagnosed with attention-deficit/hyperactivity disorder [54], and socioeconomically disadvantaged smokers [39]. The participants’ characteristics could influence the abstinence outcomes because they could have more difficulties in quitting smoking. In this vein, studies have shown that people with psychiatric diagnoses [57,58,59,60] or who are socially disadvantaged [61] have more difficulties in quitting smoking. Secondly, the heterogeneity of the inclusion criteria could influence abstinence outcomes. Some studies defined daily smokers as smoking at least 10 cigarettes/day during no less than a year e.g., [40,45], while others defined daily smokers as smoking at least five cigarettes/day for a minimum of 3 months [51]. Both CPD and years of smoking are related to nicotine dependence severity [62], which is associated with greater difficulties to achieve [63,64] and maintain abstinence [65]. Thirdly, some studies were initial evaluations of the apps’ interventions e.g., [42,43], and most were pilot e.g., [37,41,50,52] or feasibility studies e.g., [35,36,39,48,52,54], with small sample sizes, ranging from 3 to 89 participants [54,55] (Appendix A). Therefore, these studies may be underpowered to detect significant differences between conditions. These findings are congruent with the literature suggesting that smoking cessation apps are promising strategies to quit, but that the methodological limitations of studies preclude establishing their efficacy [66].

When analyzing the advantages of different kinds of apps, FFSC-Apps could increase the intensity of the smoking cessation treatments because combining an app with a face-to-face treatment offers more tools to quit. The present findings did not allow the confirmation that FFSC-Apps improve abstinence outcomes. A plausible explanation is that this category includes studies in which face-to-face contact is defined as at least one visit to the laboratory or to the smoking cessation service. Therefore, the characteristics of the studies included in the FFSC-Apps category are quite different. For instance, Minami et al. [52] included two in-person counseling and two brief phone sessions, while Masaki et al. [50] included five face-to-face sessions following the smoking cessation standard protocol in Japan.

Regarding GSC-Apps advantages, these kinds of apps can reach more people, increasing the number of people who have access to smoking cessation treatments because they are offered anywhere and at any time. Thus, people who cannot access a face-to-face smoking cessation treatment (e.g., hospital) can stop smoking through a mobile app. In addition, having a large number of people receiving smoking cessation treatment could be related to an increase in the number of people who quit smoking. Therefore, considering all of the above arguments, both kinds of apps could play an important role in the smoking cessation field.

Regarding smoking cessation apps’ features, in most studies, apps are scarcely described, and detailed information is not provided. This finding is in line with the review conducted by Vilardaga et al. [16], which also highlights the lack of complete information about smoking cessation apps.

This review has some limitations: (1) Only a few studies included a 6-month or longer follow-up or point-assessment e.g., [32,36,45,46,47,50,55], which limits the examination of the long-term sustainability of the treatment effects. (2) Most of the studies were carried out in the United States and only five in other countries [32,34,45,46,50], which should be considered within the interpretation and generalization of results. Future studies are needed to examine the effects of smoking cessation apps in other geographical and cultural settings. (3) Apps featuring comparisons are challenging because the descriptions and definitions of the components are limited. In this line, Vilardaga et al. [16] underlined the relevance of creating app design guidelines. In addition, studies have not clarified if some components of the smoking cessation apps are intended to be a therapeutic component or just for research purposes. For instance, CO measurement could be used as a tool to verify abstinence biochemically, providing a reliable research outcome, or it could be used to provide feedback to the smokers about their progress. (4) Most studies have not indicated the end of treatment or differentiated between different point-assessments in terms of app usage. This limits the analysis of tobacco use outcomes, making it difficult to establish comparisons between studies. (5) Of the 24 studies included in this review, thirteen used an intention-to-treat approach e.g., [32,40,47]. This could be affecting the results found because some studies only considered data from participants who completed the treatment, and treatment completion or higher session attendance is associated with a higher likelihood of quitting [67,68]. (6) Most of the studies did not provide information about treatment adherence, and participants had the app available during the entire study. Future investigations are needed that include information about app usability by participants and follow-up assessments once the app is not available. (7) Finally, the abstinence outcome criteria used in the studies were varied. For instance, some studies used continuous abstinence [50], whereas others used 7-day point-prevalence abstinence [52].

Despite the limitations, this systematic review has some strengths. This study describes and follows the international PRISMA statement [27] and overviews the available research about smoking cessation apps. To our knowledge, this is the first review that differentiates between GSC-Apps and FFSC-Apps, which adds to the existing literature more specific and detailed information. In addition, this is a novel and relevant research field growing because mHealth apps may significantly and positively impact population health [69].

Even though results did not support the differential effectiveness of apps to quit smoking, some considerations should be taken in favor of their use. First, mobile apps are attractive to a part of the population due to them being novel tools, which could attract more interest from people. Second, they allow access to information in an easy and fast way. Finally, as some research has suggested that smoking cessation support by mobile phone messages is cost-saving [70,71], the use of smartphone apps could be a cost-effective approach to quit smoking.

In summary, smoking cessation apps are promising tools that could be easily integrated into smoking cessation treatments. They may be able to improve some clinical aspects such as motivation and treatment adherence. Moreover, professionals can use these apps to facilitate communication with the patient, provide content in an easier way, and obtain different data that can improve the effectiveness of treatments.

## 5. Conclusions

These findings suggest that smoking cessation smartphone apps could be promising tools for smoking cessation. The mHealth apps can be considered as one more tool to quit smoking that can complement established conventional cessation treatments.

More research with strong methodological quality is needed to determine more accurately the effect of mobile apps, combined or not with face-to-face contact, on smoking cessation outcomes. Moreover, future studies should design smoking cessation apps adhering to standard guidelines [72,73] and using rigorous methodologies, including sample size calculations, intention-to-treat analysis, and longer follow-up periods. Due to the emerging development of this field, it is expected that future research will resolve the current limitations to draw clear conclusions.

## Figures and Tables

**Figure 1 ijerph-18-11664-f001:**
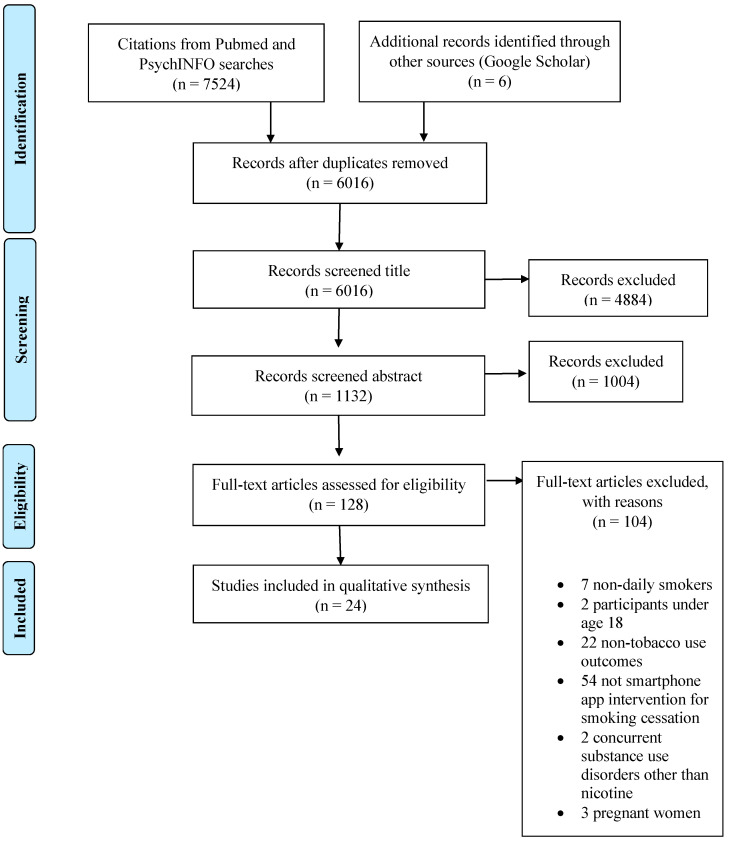
PRISMA flowchart.

**Figure 2 ijerph-18-11664-f002:**
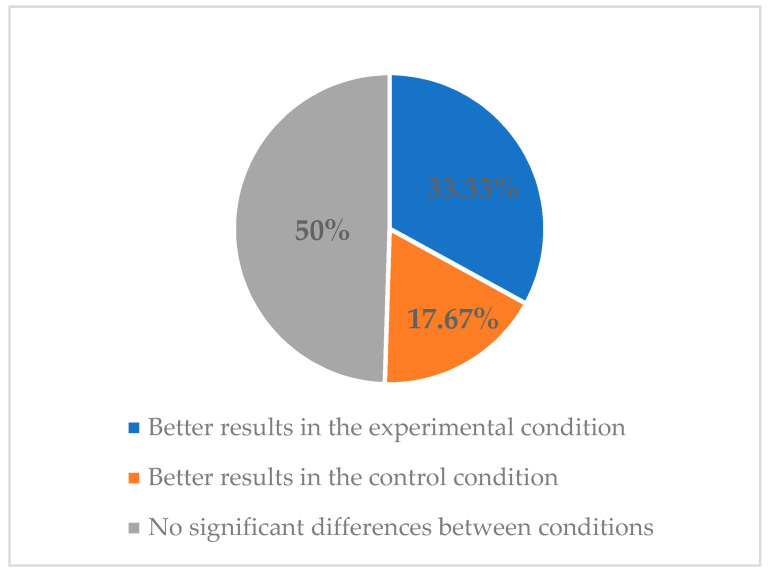
Classification of articles with strong methodological quality based on the abstinence results obtained.

**Figure 3 ijerph-18-11664-f003:**
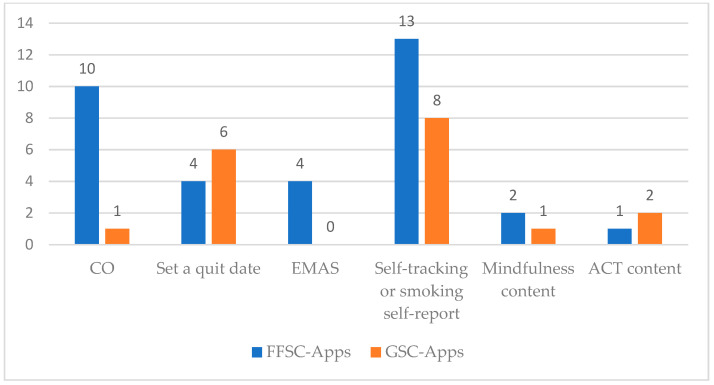
Features of FFSC-Apps and GSC-Apps. CO carbon monoxide, EMAS ecological momentary assessment, ACT acceptance and commitment therapy, FFSC-Apps smoking cessation smartphone apps combined with face-to-face intervention, GSC-Apps general smartphone apps for smoking cessation without face-to-face contact.

**Table 2 ijerph-18-11664-t002:** Main smoking cessation outcomes according to apps classification.

**General Smartphone Apps**			
Author	Abstinence	Tobacco Use	Relapse Rates
Baskerville et al. (2018) [55]	Significant differences between conditions at 6 months point-assessment in favor of the control group (22.3% vs. 18.3%).	Nonsignificant differences in CPD at 6-month point-assessment between conditions.	Not reported.
BinDhim et al. (2018) [32]	Significant differences between conditions in continuous abstinence rates at 10-days (32.2% vs. 20.8%), 1- (28.5% vs. 16.9%), 3- (23.8% vs. 10.2), and 6-month point-assessments (10.2% vs. 4.8%) in favor of the experimental group.	Not reported.	Not reported.
Bricker et al. (2014) [33]	Nonsignificant differences between conditions at 2-month post-enrollment point-assessment (13% experimental vs. 8% control).	Not reported.	Not reported.
Bricker et al. (2017) [38]	21% for 7-day PPA and 11% for 30-day PPA at 2-month-post-enrollment point-assessment.	75% reduction rate of CPD at 2-month point-assessment.	Not reported.
Buller et al. (2014) [44]	Significant differences between conditions at 6-week point-assessment in favor of the control group (58% vs. 30%).	Not reported.	Not reported.
Dar (2017) [34]	Not reported.	Significant differences in CPD reduction between conditions in favor of the experimental group at the end of the study.	Not reported.
Garrison et al. (2020) [47]	Nonsignificant differences between conditions at 6-month point-assessment (9.8% experimental vs. 12.1% control).	Nonsignificant differences between conditions in CPD reduction. Significant reductions in CPD from baseline to the 6-month point-assessment.	Not reported.
Iacoviello et al. (2017) [43]	45.2% for 7-day PPA and 26.2% for 30-day PPA at the end of the study.	Not reported.	Not reported.
Marler et al. (2019) [42]	32.0% for 7-day PPA and 27.6% for 30-day PPA at the end of the study.	Nonabstinent participants reduced 29.1% in CPD at the end of the study.	Not reported.
**Combine apps with face-to-face contact**			
**Author**	**Abstinence**	**Tobacco use**	**Relapse rates**
Businelle et al. (2016) [39]	41% at quit date, 17% at 1-week, 31% at 2-week, 27% at 3-week, 22% at 4-week, and 20% at 12-week point-assessment.	Not reported.	Not reported.
Carpenter et al. (2015) [40]	50% at 4 weeks. Of these, 65% at 3-months and 60% at 6-months point-assessment remained abstinent.	Not reported.	Not reported.
Dan et al. (2016) [54]	3% at baseline, 42% at tapering, 55% at treatment, and 42% at thinning. 0% at 1-week follow-up were abstinent.	Not reported.	Not reported.
Hébert et al. (2020) [41]	Nonsignificant differences between conditions. 22% Smart-T2, 26% QuitGuide, 30% usual care at 4 weeks point-assessment. 22% Smart-T2, 15% QuitGuide, 15% usual care at 12-weeks point-assessment.	Not reported.	Not reported.
Hertzberg et al. (2013) [35]	Nonsignificant differences between conditions at 4-week point-assessment (82% experimental vs. 45% control).	Not reported.	Not reported.
Hicks et al. (2017) [36]	Nonsignificant differences between conditions at post-treatment (60% experimental vs. 100% control) and at 2-week point-assessment (60% experimental vs. 67% control).	Not reported.	Not reported.
Janes et al. (2019) [49]	Not reported.	Nonsignificant differences between conditions in CPD reduction. Significant reductions in CPD from baseline to 1-month follow-up under both conditions.	Not reported.
Krishnan et al. (2019) [37]	Nonsignificant differences between conditions at 30-day point-assessment (3% experimental vs. 2% control).	Nonsignificant differences in CPD between baseline and 30-day point-assessment.	Not reported.
Masaki et al. (2019) [50]	64% from weeks 9 to 24, 76% from weeks 9 to 12, and 58% from 9 to 52 weeks in continuous abstinence rate.	Not reported.	Not reported.
Masaki et al. (2020) [46]	Significant differences between conditions in continuous abstinence rates from weeks 9 to 12 (75.4% vs. 66.2%), 9 to 24 (63.9% vs. 50.5%), and 9 to 52 (52.3% vs. 41.5%) in favor of the experimental group.	Not reported.	Significant differences between conditions in time to the first lapse after the quit date in favor of the experimental group.
McClure et al. (2018) [51]	25% at the quit date and 0% at the 5-day follow-up.	Not reported.	Not reported.
Minami et al. (2018) [52]	12.5% at 2-week, 4-week, and 3-months point-assessment.	All participants reported reductions in CPD from baseline to 2-week, 4-week, and 3-month point-assessments.	Not reported.
O’Connor et al. (2020) [45]	Nonsignificant differences between conditions at post-treatment (36% combined group, 20% ACT, and 24% behavioral support). Nonsignificant differences between conditions at 6-month follow-up (24% combined group, 24% ACT, and 20% behavioral support).	Significant differences in CPD reduction in favor of the combined condition.	Not reported.
Raiff et al. (2017) [48]	1.25% at baseline, 13.8% at tapering, 35.5% at abstinence induction, and 0% at 1-month follow-up.	Not reported.	Not reported.
Wilson et al. (2019) [53]	Cohort 1: 40% at post-treatment and 20% at 3-months follow-up. Cohort 2: 38% at post-treatment and 15% at 3-months follow-up.	Cohort 1: 20% reduced CPD at post-treatment. Cohort 2: 38% reduced CPD at post-treatment	Not reported.

ACT acceptance and commitment therapy, App smartphone application, CPD cigarettes per day, PPA point prevalence abstinence.

**Table 3 ijerph-18-11664-t003:** Definitions of features of smartphone apps for smoking cessation.

Feature	Definition
CO	Taking a CO breath sample
Set a quit date	Creating a tailored quit plan or set a quit date
EMAS	Ecological momentary assessment, whose definition appears in each study
Self-tracking or smoking self-report	Providing information about self-progress through the smoking cessation process or logging cigarettes in the app.
Mindfulness content	Any information, material, or activity based on mindfulness
ACT content	Any information, material, or activity based on acceptance and commitment therapy

CO carbon monoxide, EMAS ecological momentary assessment, ACT acceptance and commitment therapy.

**Table 4 ijerph-18-11664-t004:** Features of the apps included in the studies.

Smartphone App	CO	Set a Quit Date	EMAS	Self-Tracking or Smoking Self-Report	Mindfulness Content	ACT Content
CTC app [55]		X		X		
Intervention app [32]		X		X		
SmartQuit [33,45]		X		X		X
SQ2.0 app [38]		X		X		X
REQ-Mobile [44]		X				
SmokeBeat app [34]				X		
Craving to Quit [47]				X	X	
Clickotine app [43]		X		X		
Pivot mobile app [42]	X			X		
Smart-T app [39]			X			
mCM app [35,36,40,53]	X			X		
Motiv^8^ app [48,54]	X			X		
Smart-T2 app [41]		X	X			
Stay Quit Coach app [36,53]				X		
App-based MT program [49]		X		X	X	
Coach2Quit app [37]	X	X		X		
CASC smartphone app [50]				X		
CASC smartphone app [46]	X			X		
M3 app [51]	X		X	X		
Smartphone app [52]	X		X	X	X	

App smartphone application, CO carbon monoxide, EMAS ecological momentary assessment, ACT acceptance and commitment therapy.

## Data Availability

The data supporting the conclusions of this article are included within the manuscript and its Appendix A.

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
