# Peer review of "Smoking Cessation Apps: A Systematic Review of Format, Outcomes, and Features"

_ijerph, 2021, doi:10.3390/ijerph182111664_

Round 1

Reviewer 1 Report

Dear Authors,

My concerns were addressed appropriately. 

Author Response

Dear Authors,

My concerns were addressed appropriately. 

We thank the Reviewer for his/her favorable comment on the study.

Reviewer 2 Report

I have read with a lot of interest the work entitled "Smoking Cessation Apps: a Systematic Review of Format, Outcomes, and Features" by Barroso-Hurtado et al. I should recognize that the work provides interesting insights about the value of the Smoking Cessation Apps and their needed characteristics. However, there several minor points to adjust prior to its publication: * The Methods include that six additional records were identified through other sources. Authors are kindly invited to mention these other data sources. * The queries used to extract records from Pubmed and PsychINFO are not involved in the Paper. Authors should involve these queries (e.g. "Smoking" AND "Cessation" AND "App") to ensure the reproducibility of the systematic review. * A definition of Abstrackr is needed. Based on these points, we invite the editors to accept this systematic review for publication after these minor revisions are applied to the Paper.

Author Response

I have read with a lot of interest the work entitled "Smoking Cessation Apps: a Systematic Review of Format, Outcomes, and Features" by Barroso-Hurtado et al. I should recognize that the work provides interesting insights about the value of the Smoking Cessation Apps and their needed characteristics. However, there several minor points to adjust prior to its publication:

We thank the Reviewer for the comments on the study, which have helped us to improve the manuscript

  1. The Methods include that six additional records were identified through other sources. Authors are kindly invited to mention these other data sources.

Thank you for your comment. We have clarified this information in the PRISMA flowchart (page 4). In addition, in section "2.1. Search strategy" we have specified that "Additionally, a search of the first 200 citations published online in Google Scholar was undertaken.” (page 2, paragraph 4, lines 93-94).

  1. The queries used to extract records from Pubmed and PsychINFO are not involved in the Paper. Authors should involve these queries (e.g. "Smoking" AND "Cessation" AND "App") to ensure the reproducibility of the systematic review.

We appreciate this comment. Following the reviewer recommendation, we have added the Table S1 that contains this information. Also, the following sentence has been added to the text: “Databases searched were Pubmed and PsycINFO. The complete literature search strategy can be found in Table S1.” (page 2, paragraph 4, lines 92-93).

  1. A definition of Abstrackr is needed.

Following the reviewer suggestion, we have included a brief description of Abstrackr in page 3, paragraph 7, lines 132-134: “This is a free tool to upload and organize the results obtained in systematic reviews, al-lowing filter and organize the abstracts of the articles on a single platform”.

Based on these points, we invite the editors to accept this systematic review for publication after these minor revisions are applied to the Paper.

We appreciate all the Reviewer comments and suggestions for improvement of the manuscript.

Reviewer 3 Report

Highly significant manuscript in the context of wide utilization of smartphone applications. The present study identified the scarcity of studies conducted to assess the quality, content and impact of smoking cessation interventions on smoking behaviors, relapse and cessation.  The study makes important contribution to the literature including a strong methodology to review the literature including the use of a score to classify the quality of the methodological quality of the applicative (strong, moderate or weak). Overall, the present manuscript is well organized but could benefit from additional review to better guide review. In the purpose to shorten the manuscript, authors should carefully consider deleting some figure/tables that could be easily described.

3.1. Results should include % to facilitate reader understanding of the quality / types of applications.

While table 1 is informative, authors should consider adding a table to describe the effect of the apps (with facetime and general) on cessation, tobacco use and relapse to facilitate readers comprehension of the main findings.  Table 1 could be described, and a new table should be constructed to report 3.3. findings.

Figure 2 can be easily described in the text saving space and improving clarity.

Table 2 could be described in the text as you describe figure 3. 

Discussion is excessively long.  The discussion section should be revised and summarized accordingly the main purpose of the paper.  

Author Response

Highly significant manuscript in the context of wide utilization of smartphone applications. The present study identified the scarcity of studies conducted to assess the quality, content and impact of smoking cessation interventions on smoking behaviors, relapse and cessation.  The study makes important contribution to the literature including a strong methodology to review the literature including the use of a score to classify the quality of the methodological quality of the applicative (strong, moderate or weak). Overall, the present manuscript is well organized but could benefit from additional review to better guide review. In the purpose to shorten the manuscript, authors should carefully consider deleting some figure/tables that could be easily described.

We thank the Reviewer for the positive comments about the present study. The suggestions provided have helped us to improve the quality of the manuscript. Please, find our point-by-point answers as follow.

  1. Results should include % to facilitate reader understanding of the quality / types of applications.

We appreciate this comment. Following the reviewer suggestion, we have added the following percentages (Page 5, paragraph 3, lines 175-185): “The characteristics of the studies included are described in detail in Table S21. Of the 24 included studies, 19 (79.1%) were conducted in the United States, two (8.3%) in Japan, one (4.2%) in Israel, one (4.2%) in Ireland, and one (4.2%) in Australia. Regarding the study type, nine (37.5%) were RCTs [e.g.,32,33], four (16.7%) were controlled clinical trials (CCTs) [34-37], and eleven (45.8%) were before-and-after [e.g.,38,39]. Fifteen (62.5%) used FFSC-Apps [e.g.,39,40] and nine (37.5%) used GSC-Apps [e.g.,32,33]. Regarding the studies with FFSC-Apps, two (13.3%) of these studies were pilot [37,42] and two (13.3%) were fea-sibility studies [35,36]. Regarding GSC-Apps, two (22.2%) of these studies were initial evaluations [47,49]. Fifteen studies (62.5%) included intervention assessment points ranging from 1 to 52 weeks, and four studies (16.7%) included follow-up assessments ranging from post-treatment to 6-months.”

  1. While table 1 is informative, authors should consider adding a table to describe the effect of the apps (with facetime and general) on cessation, tobacco use and relapse to facilitate readers comprehension of the main findings.  Table 1 could be described, and a new table should be constructed to report 3.3. findings.

Thank you for your suggestion, this is an interesting question. Following the reviewer's recommendations, a new table has been included to facilitate the understanding of “3.3. Effects of smartphone apps on abstinence, tobacco use, and relapse rates” section (page 7, lines 248-250). Due to the inclusion of  this new table, the text of this section has been reduced (page 7-11, lines 207-297).

  1. Figure 2 can be easily described in the text saving space and improving clarity.

We appreciate the reviewer comment. We have made changes in the results section in order to improve clarity (page 7-11, lines 207-297).

  1. Table 2 could be described in the text as you describe figure 3. 

Thank you for your comment. As recommended, we have included a description of table 2 in the text (Page 11, paragraph 5, lines 317-319): “Regarding the features of smoking cessation apps (Table 3), we have clustered them into the following groups depending on the content of the apps: CO; set a quit date; EMAS; self-tracking or smoking self-report; mindfulness content; and ACT content.”

  1. Discussion is excessively long.  The discussion section should be revised and summarized accordingly the main purpose of the paper.

Thank you for your suggestion. Now, the discussion has been reduced and summarized as recommended (Page 14-16, lines 378-454).

Reviewer 4 Report

The authors of the paper “Smoking Cessation Apps: a Systematic Review of Format, Outcomes, and Features” performed a systematic review on existing mobile apps in the market for smoking cessations. They made a distinction between apps used in conjunction with face-to-face intervention and apps used alone, something different from existing reports in the literature. The reported their findings in accordance with the PRIMSA format, which give its reader sufficient information to assess the quality of their work. Key findings include smoking cessation apps are useful for smokers who want to quit and better-quality data are needed for more definitive conclusions about the effectiveness of smoking cessation apps. The authors set the context of the study and gap of knowledge in a compelling and informative way.

Some suggested areas for improvement:

  1. Abstract: Please refrain from using abbreviations (e.g., CO) without spelling it in full at first instance.
  2. Materials and Methods: Please provide list of search terms used so that reviewers and readers may attempt to replicate study, if needed. Recommend to report search terms in PICO format (see https://www.ncbi.nlm.nih.gov/pmc/articles/PMC6148624/).
  3. Results:
    1. I am keen to know if there is a correlation between the ratings of methodological quality by the EPHHP tool and the outcomes of the study – can you do a table to chart information from Table 1 and compare with the outcome data from Figure 2?
    2. I wonder if the authors are open to run a Relative Risk analysis on the data from studies included in this systematic review. For example, https://pubmed.ncbi.nlm.nih.gov/30349201/, for more info.
  4. Discussion: For authors to consider including some discussion on implications for professionals working with people who want to quit smoking—how does the conclusions from this study contribute to clinical practice?

Author Response

The authors of the paper “Smoking Cessation Apps: a Systematic Review of Format, Outcomes, and Features” performed a systematic review on existing mobile apps in the market for smoking cessations. They made a distinction between apps used in conjunction with face-to-face intervention and apps used alone, something different from existing reports in the literature. The reported their findings in accordance with the PRIMSA format, which give its reader sufficient information to assess the quality of their work. Key findings include smoking cessation apps are useful for smokers who want to quit and better-quality data are needed for more definitive conclusions about the effectiveness of smoking cessation apps. The authors set the context of the study and gap of knowledge in a compelling and informative way.

We want to thank the reviewer for the insightful comments and suggestions regarding this study. We have incorporated the reviewer's recommendations into the manuscript, which can be found in the answers below.

Some suggested areas for improvement:

  1. Abstract: Please refrain from using abbreviations (e.g., CO) without spelling it in full at first instance.

Thank you for your recommendation. We have checked the abstract and full-text of the study for abbreviations (e.g., CO) without spelling. We have included the complete word before the abbreviation (page 1, line 20): “carbon monoxide (CO) measures. Smartphone apps for smoking cessation could be promising tools.”.

  1. Materials and Methods: Please provide list of search terms used so that reviewers and readers may attempt to replicate study, if needed. Recommend to report search terms in PICO format (see https://www.ncbi.nlm.nih.gov/pmc/articles/PMC6148624/).

We appreciate this comment. Following the reviewer recommendation, we have added the Table S1 that contains this information. Additionally, the following sentence has been added to the text: “Databases searched were Pubmed and PsycINFO. The complete literature search strategy can be found in Table S1.” (page 2, paragraph 4, lines 92-93).

  1. Results: I am keen to know if there is a correlation between the ratings of methodological quality by the EPHHP tool and the outcomes of the study – can you do a table to chart information from Table 1 and compare with the outcome data from Figure 2?

Thank you for this comment. We agree the reviewer that a correlation between the ratings of methodological quality and the outcomes of the study would be interesting. Unfortunately, due to the hetereogenity of the results (i.e., some studies report continuous abstinence, while others use 7- or 30-days point prevalence abstinence) and methodology designs (i.e., some studies have a before and after design and have not a comparison condition) of the included studies, it is not feasible to conduct this analysis.

  1. I wonder if the authors are open to run a Relative Risk analysis on the data from studies included in this systematic review. For example, https://pubmed.ncbi.nlm.nih.gov/30349201/, for more info.

Thanks for this comment, since running a Relative Risk analysis could be very interesting and informative. Unfortunately, conducting this type of analysis is not feasible for the following reasons: 1) studies included in the present systematic review have measuredthe outcome variables in different ways (for instance, some studies used 7-days point prevalence abstinence while others used prolongued abstinence); 2) there is a lot of variation between the results of the included studies (heterogeneity); and 3) a large amount of studies were initial evaluations, pilot and feasibility studies of the apps interventions (González et al., 2011; Haidich, 2010). We hope that future research will allow conducting this type of analysis.

Ferreira González, I., Urrútia, G., & Alonso-Coello, P. (2011). Systematic reviews and meta-analysis: scientific rationale and interpretation. Revista Española de Cardiologia, 64(8), 688–696. https://doi.org/10.1016/j.recesp.2011.03.029

Haidich A. B. (2010). Meta-analysis in medical research. Hippokratia, 14(Suppl 1), 29–37.

  1. Discussion: For authors to consider including some discussion on implications for professionals working with people who want to quit smoking—how does the conclusions from this study contribute to clinical practice?

Thank you for your comment. Following the reviewer suggestion, we have provided some information regarding implications for smoking cessation professionals (Page 15-16, paragraph 4, lines 471-475): “In summary, smoking cessation apps are  promising tools that could be easily inte-grated into smoking cessation treatments. They may be able to improve some clinical as-pects such as motivation and treatment adherence. Moreover, professionals can use these apps to facilitate communication with the patient, provide content in an easier way and obtain different data that can improve the effectiveness of treatments”.

This manuscript is a resubmission of an earlier submission. The following is a list of the peer review reports and author responses from that submission.

Round 1

Reviewer 1 Report

The article is of interest but requires a very thorough review of the work before it can be considered for publication. The reviewer congratulates the authors for their work. However, the reviewer recommends the rejection of the work, recommending to apply the following guidelines for the improvement of the work and its re-submission back to the journal once it has been matured.

The review methodology is adequate.

In relation to the results section:

The final number of Smoking Cessation Apps is small enough (24) to include a new section in the results that summarizes the characteristics of the different applications. In the opinion of the reviewer, this new section would provide a higher quality to the review work, and would be of interest to the readers. In addition, it would serve to highlight the characteristics of common applications for Smoking Cessation. One or two illustrative images of the common features of the applications would be very suitable

The results obtained from the review are explained in a very systematic and structured way. However, a deep comparative analysis of the results obtained by the different authors is lacking. The results presented are confusing, largely due to the disparity of results obtained by the different authors. However, an effort should be made in writing so that meaningful conclusions can be drawn from the review study.

Comparative results diagrams and figures would improve the readability of the results, and allow for better comparative analysis. The reviewer recommends making an effort to be able to compare the results of the works using figures, tables and diagrams.

Table 1 shows a Ratings of methodological quality by the EPHHP tool. This table would make more sense if the results of methodological quality are taken into account when comparing the results of the different authors, to give greater credibility to the most complete studies from the methodological point of view. These aspects should be included when analyzing the results in a comparative way.

In relation to the discussion section:

In the opinion of the reviewer, the discussion section should start from the conclusions of the results obtained in the comparative analysis of the results previously carried out, and from there, highlight these results with respect to the bibliography, either to provide an explanation to the results, to corroborate or refute results, or even to show new findings. The discussion section is confusing, in part because it mixes with results, and does not altogether follow the lines that a discussion should have.

In relation to the conclusion section:

The conclusions of the work are not clear, largely due to the deficiencies described in the discussion and conclusion section. The reviewer recommends redoing the results and discussion sections following the recommendations previously described. Surely then less confusing conclusions can be drawn. The disparity of results should not be a limit to draw conclusions from the study. Although it should be noted as limitations that most of the studies are pilot studies or feasibility studies, it would be desirable to provide clear conclusions from the review work.

Reviewer 2 Report

Dear Authors,

Comments and suggestions for the manuscript “Smoking Cessation Apps: a Systematic Review of Format, Outcomes, and Features”

The manuscript describes the effects of FFSC-Apps and GSC-Apps in smoking cessation interventions.

  1. The introduction talks about the smoking apps, but there is no description of the type of apps. Are these smartwatch apps, smartphone apps, or other types of wearable devices that record these interventions?
  2. What is PROSPERO in line 89?
  3. How are the intervention integrity and analysis appropriate measured in lines 175-176?
  4. In line 181, what does “appropriate statistical methods” mean?
  5. In lines 190-191, what does it mean “app to a wait-list control condition” and “OnRQ” ?
  6. Could you please provide some details/information regarding the intervention app and the control app (in lines 196-197)?
  7. What is PPA in line 200?
  8. What is CASC group in line 230?
  9. What is ACT face-to-face treatment in line 235?
  10. In line 291, how small are the “small sample sizes”?
  11. What is “an active control condition” in line 293?